# Results of Aortic Coarctation Repair in Low- and Normal Birth-Weight Neonates: A Propensity Score-Matched Analysis

**DOI:** 10.3390/life13122282

**Published:** 2023-11-29

**Authors:** Aleksandra Krylova, Andrey Svobodov, Margarita Tumanyan, Elena Levchenko, Sergey Kotov, Yuliya Butrim, Vladimir Shvartz

**Affiliations:** 1The Department of Intensive Cardiology for Congenital Heart Diseases of Premature Neonates and Infants, Bakulev National Medical Research Center for Cardiovascular Surgery, Moscow 121552, Russia; 2The Department of Surgical Treatment for Interactive Pathology, Bakulev National Medical Research Center for Cardiovascular Surgery, Moscow 121552, Russia

**Keywords:** aortic coarctation, congenital heart disease, low birth weight, prematurity

## Abstract

Introduction: Surgical treatment of aortic coarctation in low-birth-weight (LBW) neonates is associated with risks of higher hospital mortality rates and recoarctation development in the long-term. The goal of our study was to compare the results of surgical treatment of aortic coarctation in LBW neonates and normal-weight patients, to identify predictors of recoarctation in the long-term among LBW patients. Materials and methods: A retrospective study was performed to analyse the patients who had undergone resection of isolated aortic coarctation between 2005 and 2022. Overall analysis included 521 patients under the age of 30 days, 89 LBW patients and 432 patients with normal body weight. Propensity score matching (PSM) was performed at a ratio of 1:1 for the statistical adjustment of original patients’ data in both groups. Results: All patients underwent standard resection of aortic coarctation with extended end-to-end anastomosis. Hospital mortality among LBW patients was 8% and in patients with normal weight the mortality amounted to 1% (*p* = 0.030). LBW patients were transferred to other hospitals more often than normal-weight patients who were more often discharged (*p* < 0.001). In the long-term, period mortality was not statistically significantly different (*p* = 0.801). The freedom from reoperation in the group of normal-weight children was 87%, whereas in the LBW patients the rate was 63% (log rank test, *p* = 0.104). In the multivariate regression model, the most significant risk factors for reoperation were as follows: preoperative inotropes administration (OR (95% CI) 4.369 (1.316–14.51)) and pressure gradient across aortic arch before discharge (OR (95% CI) 1.081 (1.014–1.153)). Conclusions: Hospital mortality was higher among LBW patients (*p* = 0.030). There was a statistical trend of differences in reoperation rates: in the long-term among LBW patients, a higher reintervention probability remains. Moreover, the LBW group initially had more severe clinical condition in terms of cardiac failure and impaired renal function. Factors associated with the risk of recoarctation were preoperative inotropes infusion and pressure gradient across aortic before discharge.

## 1. Introduction

Coarctation of the aorta is a congenital heart defect (CHD) characterised by a haemodynamically significant obstruction to antegrade blood flow in aortic isthmus. Aortic coarctation occurs in 2–5 out of 10,000 newborns and accounts for 6–7% of all congenital heart defects [1]. The birth of low-birth-weight (LBW) infants as a result of preterm labour or intrauterine growth restriction is a current issue in obstetrics and neonatology. Among all neonates, the percentage of those with a LBW ranges from 5 to 16%, according to World Health Organization (WHO) (2012) [2].

Over past decades, accumulated experience has improved the results of cardiac surgeries in neonates, but LBW still continues to be a significant risk factor for mortality among patients with CHD [3,4,5]. LBW and premature infants have various comorbidities, such as bronchopulmonary dysplasia, necrotising enterocolitis, intraventricular haemorrhage, coagulopathies, and immaturity of organs, which aggravate the course of the perioperative period. Some authors recommend surgery instead of a wait-and-see approach to achieve somatic growth [6,7]. The most optimal surgical correction of these patients is coarctation resection with end-to-end anastomosis. According to some authors, surgical treatment in LBW neonates is associated with higher risk of recoarctation and hospital mortality [8,9]. However, other studies have not revealed any correlation between low weight, recoarctation, and survival rates [6,7].

The aim of this study Is to compare the results of surgical treatment of aortic coarctation in LBW and normal weight neonates and to identify predictors of recoarctation among LBW patients.

## 2. Materials and Methods

### 2.1. Study Groups

Patients who underwent aortic coarctation repair between 2005 and 2022 were analysed retrospectively. Low-birth-weight neonates were defined as those with body weight less than 2500 g, premature infants were defined as those born between 22 and 37 weeks of gestation, according to WHO (2012). Inclusion criteria for the study group were as follows: weight at admission less than 2500 g, isolated coarctation of the aorta, neonates. The exclusion criteria were as follows: presence of other haemodynamically significant CHD, age older than 30 days. As a result, 89 patients were selected and formed the study group. The control group included 432 patients with isolated coarctation of the aorta who weighed more than 2500 g at admission.

### 2.2. Data Collection

All patients were examined, including review of patients’ histories, physical examination, grading the heart failure according to modified Ross classification, as well as echocardiographic assessment of left heart structures, patent ductus arteriosus (PDA) diameter, distal aortic arch diameter, isthmus and pressure gradient across it. Coarctation was diagnosed by echocardiography, patients with hypoplastic aortic arch or unusual anatomy underwent CT scan before surgery. Postoperative aortic arch gradient was measured twice—on the first day after surgery and before discharge. Aside from standard laboratory tests, the following additional parameters were taken into account: serum lactate level before and after surgery, creatinine level, glomerular filtration rate (GFR) calculated according to the Schwartz formula [10], and the patient’s pRIFLE score, which is a modified scale for assessing the acute kidney injury in children [11]. The vasoactive-inotropic score during the first 24 h after surgery was calculated to assess the mortality risk using the stratification system offered by European Association of Cardiothoracic Surgeons [12]. All data were collected from the electronic database the “MedWork” system in clinic in compliance with all legal guidelines.

### 2.3. Endpoints

The primary endpoint was in-hospital mortality and long-term survival. The secondary endpoint was the development of recoarctation in the long-term.

### 2.4. Surgery

All patients underwent resection of aortic coarctation with extended end-to-end anastomosis. The surgery was performed from the left posterolateral thoracotomy through the third intercostal space, without bypass. The PDA was ligated, the aorta was clamped proximally and distal to the coarctation site, the coarctation with ductal tissue was excised, and an extended anastomosis between the arch and descending aorta was performed using prolene 7–8/0 sutures. Indications for surgery were as follows: isthmus Z-score less than −3, pressure gradient more than 20 mmHg and symptoms of heart failure based on Ross Classification.

### 2.5. Statistical Analysis

The median and interquartile range Me value (Q1; Q3) were used to describe the parameters. The non-parametric Mann–Whitney test (for quantitative data) and Pearson’s chi-square test (for qualitative data) were used to compare two independent samples. Propensity score matching (PSM) was performed to compare the findings in our study groups. A total of 521 patients were matched at a 1:1 ratio based on propensity scores. Propensity scores were calculated for each patient using multivariable logistic regression based upon the covariates such as age, sex, presence of comorbidities (perinatal encephalopathy, neonatal jaundice, multiple congenital anomalies (MCA), history of infections) and through calipers of width equal to 0.01 of the standard deviation of the logit of the propensity score.

Log-rank test was used to compare groups by endpoints occurring over time, with Kaplan–Meier survival curves constructed. Univariate and multivariate logistic regression analyses were used to identify predictors of recoarctation. It was derived as odds ratio (OR), in the 95% confidence interval (CI), and *p*-value. The software packages used were STATISTICA 10 (Statsoft, Tulsa, OK, USA), MedCalc Version 22.016 (MedCalc Software Ltd., Ostend, Belgium), and SPSS^®^ Statistics 28.0 software (Chicago, IL, USA).

## 3. Results

### 3.1. Low-Birth-Weight vs. Normal-Weight Patients

The primary clinical characteristics of the groups and after PSM are presented in Table 1. After statistical correction, the groups differed only by anthropometric parameters (birth weight, weight at admission, height, and body surface area (BSA)) and gestational age. Statistically significant differences in the frequency of comorbidities were levelled out. The analysis of data in the groups after statistical correction is presented below.

The initial data of patients from both groups are presented in Table 2. The groups were found to be statistically significantly different in terms of Ross class of heart failure (*p* = 0.021), in GFR (*p* < 0.001) and in incidence of mechanical ventilation at admission (*p* < 0.001). The number of patients requiring inotropic support before surgery in the LBW group was two times higher than in normal-weight patients. Comparing the echocardiographic data, the groups differed in Z-score of isthmus and distal arch. The isthmus was more hypoplastic in normal-weight patients, whereas the distal arch was smaller in LBW infants.

### 3.2. Intraoperative and Postoperative Complications

The intraoperative complication rate in the LBW group was 5.6% (5 patients), while in the control group all surgeries were performed without complications. The following complications were noted: bleeding in 4 patients (4.5%), and thrombosis of the anastomosis after removing aortic clamps (1 patient—1.1%), which required the reapplication of anastomosis. When assessing renal function after surgery, all parameters were significantly different between the groups; creatinine levels were higher (*p* = 0.046), GFRs were significantly lower (*p* < 0.001) in the LBW group; according to pRIFLE score the risk, injury, and failure groups were more common among LBW patients. Also, serum lactate values after surgery were higher in the LBW group (*p* = 0.034) (Table 3).

Comparing the postoperative complications in the two groups, a greater number of infections in LBW patients is noted. The groups differed statistically in incidence of necrotising enterocolitis; among LBW patients it was more than two times higher than among the children with normal weight (*p* = 0.009). There was also a higher share of patients with acute kidney failure in the postoperative period—5.6% for LBW infants, whereas only 1% (*p* = 0.054) in the control group. The LBW children had no neurological complications after surgery, while such complications occurred in 8% of control group (2 patients had ischemic strokes confirmed by computed tomography scan). Aortic cross-clamp time in patients with neurological complications was within acceptable range (20–30 min).

### 3.3. Outcomes

In the LBW group hospital mortality was 8%, whereas among normal weight children the respective parameter amounted to 1% (*p* = 0.03). The groups also differed significantly in other outcomes of hospitalisation: patients in the control group were more likely to be discharged home (82%), while in the LBW group this number was only 25% (*p* < 0.0001). Many more LBW patients (67%) compared to the control group (17%) were transferred to other paediatric hospitals (*p* < 0.0001) due to impossibility of early extubation.

### 3.4. Long-Term Results of Surgical Treatment

We were able to collect data from 63 patients in the LBW group and 76 patients from the control group. The mean duration of follow-up was 15 years. The incidence of the primary endpoint in the long-term was not significantly different (*p* = 0.801) (Figure 1A.) During the follow-up period, two patients in the LBW group died of noncardiac causes (one and six months after discharge, respectively). In the group of normal-weight patients, two patients died three and four months after initial repair also due to noncardiac causes.

In the LBW group, recoarctation developed in 13 patients (20%), 10 patients underwent balloon dilatation, 2 patients required an on-pump patch angioplasty, and 1 patient underwent stenting. Most of the reoperations were performed within the first 6 months after the initial repair but two patients required reintervention later (one patient after 14 months, the other—after 9 years). In the normal-weight group, 9 patients (12%) required reoperation. All patients underwent balloon dilation within a year after the initial repair.

Kaplan–Meier curves revealed that the freedom from reoperation in patients with normal body weight during the entire follow-up period was 87%, while in the LBW patients the respective value amounted to only 63%. Despite the absence of statistical significance of differences in the number of reoperations (log rank test, *p* = 0.104), a statistical trend can still be noted. LBW patients are more likely to require reoperation in the long-term (Figure 1B).

Since the risk of recoarctation is higher in the LBW group and it persists for a long time, we performed single-factor and multifactor regression analyses of the LBW group to determine the factors associated with the development of recoarctation.

In the univariate regression model, we found that initial respiratory rate, preoperative inotropes infusion, and the Ross class were associated with the higher occurrence of recoarctation in the long-term. Aortic cross-clamp time was also associated with the risk of reoperation, but it was inversely related to it. In the multivariate model, preoperative inotropes assignment and pressure gradient across aortic before discharge were the most significant parameters (Table 4).

## 4. Discussion

Our research showed that patients in LBW group initially had more severe clinical conditions due to heart failure and impaired renal function. More patients in the studied group were classified as Ross class I and II (33.8% and 6.7%, respectively), while in the normal-weight group 22.5% were classified as class II and 2.2% as class III. When assessing the patients’ preoperative renal function, the LBW group was characterised by higher creatinine levels, lower GFR, and based on pRIFLE score more patients were classified as subject to risk of kidney injury. Increased heart rate and impaired renal function in the LBW group might be due to early gestational age in patients with mild symptoms of heart failure (Ross class I), but among infants with initially severe clinical condition these parameters are mostly manifestations of heart failure.

More LBW patients were admitted on mechanical ventilation and required preoperative inotropes administration. Thus, it can be noted that the clinical condition of LBW and premature patients is worse than the one of normal-weight patients due to anatomical and functional immaturity of organs and systems in general, and renal tissue, in particular. According to other authors, acute kidney injury is more frequent in this cohort [13,14], especially in children with concomitant coarctation of the aorta [15,16,17].

In our study we discovered that neurological complications after surgery (seizures, ischemic strokes) appeared more often in patients with normal weight, rather than in LBW patients. The interpretation of this phenomenon is rather contradictory. Although ischemic strokes are extremely rare after aortic coarctation repair in children, other authors claim that they occur more often in older patients with significant arterial hypertension or in the presence of cerebral vascular anomalies (Willis circle anomalies, hypoplastic vertebral arteries) [18,19]. Patients with LBW and prematurity have lower peripheral vascular resistance and are more prone to hypotension due to immaturity of peripheral vasoregulation [20], and, in our opinion, such patients better tolerate manipulations with the aortic arch during the surgery. There is also a significant difference between hospitalisation outcomes between the low-weight and normal-weight groups: LBW patients are less often discharged home and more often transferred to other hospitals, which is attributed to their need for longer hospital stay after surgery in order to ensure weight gain or prolonged ventilatory support.

Patients in both groups who were discharged home had approximately the same duration of hospitalization—about 8–10 days. Many patients were transferred to our center only for surgical treatment for 2–3 days due to legal reasons and then transferred back to continue treatment in specialised perinatal centers, so we did not analyze the length of hospital stay. We were unable to collect data on patients who transferred to other hospitals because of technical reasons, so we described the outcomes of hospitalizations: number of patients who were discharged home, transferred patients, and deaths.

In this research we obtained a significant difference in the primary endpoint (hospital mortality) between the studied groups. For LBW patients after surgical treatment, the respective share was 8%, compared to 1% in the group of normal-weight infants. Similar data were obtained in research by Burch et al. (2009), in which low body weight was studied as a risk factor for mortality after surgical treatment of coarctation. The authors demonstrated hospital mortality rates of only 3.4% in children with body weight less than 2.5 kg [6]. In another study, 24 children weighing less than 2 kg after coarctation repair were followed up, the authors obtained a rather high hospital mortality rate of 20.1% (5 patients). The authors argue that the rate is associated with a severe initial condition, complex CHD and noncardiac causes that aggravate the prognosis [21]. Curzon et al. show the mortality rate of 7,1% after the coarctation repair in LBW patients, authors reveal that weight less than 2.5 kg influences the mortality [22].

In our study, recoarctation developed in 20% of patients in the LBW group, usually within 6 months after initial repair. The development of recoarctation was not influenced by weight, age and prematurity, but by parameters assessing the initial state of heart failure, such as heart rate, preoperative inotropes administration, Ross class, as well as aortic cross-clamping time and arch pressure gradient before discharge. The most significant were the gradient before discharge and preoperative inotropes administration. Several other authors also mentioned the association between higher pressure gradient across aortic arch after repair and recoarctation development in the long-term, expectedly [23,24,25].

Interesting data were obtained in a recent study by Lehnert et al. The authors claim that the higher risk for aortic arch re-intervention was associated with lower age at the time of surgery (less than 15 days) and the need for PGE1 infusion preoperatively [26]. Another research paper by Soynov et al. (2017) also describes the outcomes of surgery in patients with coarctation of the aorta. When Cox regression analysis was performed, it appeared that the only significant risk factor for the development of recoarctation was low weight at the time of surgery [9]. A great deal of authors claims that hypoplastic aortic arch is a risk factor for recoarctation development [6,7,27,28], in our study did not find the correlation.

Dr. Costopoulos K. and colleagues (2019) included only 9 patients weighing less than 2.5 kg in their analysis describing the results of surgical treatment of coarctation of the aorta. When compared with a group of normal-weight patients, the incidence of recoarctation was higher in LBW infants (8.3%), compared to 5.4% in normal-weight group, but no statistical analysis was performed [29].

As mentioned above, other studies also describe the association of recoarctation with a higher arch gradient before discharge [23,24,25,30]. However, there is insufficient data on the connection between inotropic support and recoarctation. Truong et al. (2013) also demonstrate this fact, but multivariate analysis revealed more significant factors: Z-score of the sinotubular junction, blood flow velocity in the descending aorta after surgery [31]. Analysing the effect of inotropic support on the development of recoarctation in the long term requires further study and possibly more observations.

### Limitations of the Study

This analysis had both advantages and limitations, primarily related to its retrospective approach. Firstly, despite the fact that all data were collected from the electronic database of our clinic “Medwork” with standard mandatory data entry, retrospective analysis does not exclude partial loss of information. Secondly, the use of propensity score matching does not completely exclude hidden systematic error, which could potentially affect the results. Thirdly, long-term data were not collected from all patients: 63 children in the study group and 76 patients in the control group.

We assume that the advantages of this study include a relatively large sample size including 89 LBW neonates with aortic coarctation, as well as a long follow-up period: the average duration was 15 years. For the first time in our country based on a large clinical material a comparative assessment of LBW neonates with aortic coarctation during all stages of treatment was carried out, risk factors for recoarctation were identified.

## 5. Conclusions

Over the past years the results of aortic coarctation repair in LBW and premature infants have been improving, experience with such patients has been gained, standards of medical care have been optimised, and new endovascular techniques of treatment have been improved [15,16]. Nevertheless, low-birth-weight and premature patients have higher mortality rates and are more likely to develop recoarctation in the long term [6,7,8,29].

In this study, a comparative analysis of the results of surgical treatment of coarctation of the aorta in LBW and normal-weight neonates was performed. It was found that hospital mortality and the incidence of recoarctation development are much higher in LBW patients, and that they also initially have more severe clinical conditions in terms of cardiac failure and impaired renal function.

## Figures and Tables

**Figure 1 life-13-02282-f001:**
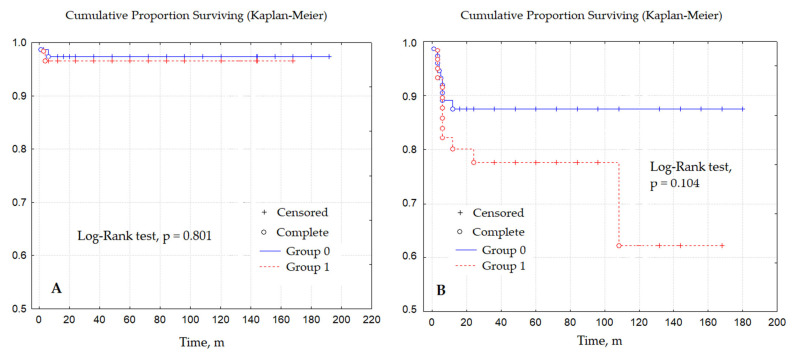
Comparison of groups in the long-term ((**A**) death from any cause; (**B**) reoperation).

**Table 1 life-13-02282-t001:** General data and comorbidities of low- and normal-birth-weight patients before and after PSM.

Parameters	Unmatched Raw Data			Propensity Matched 1:1		
	Low-Birth-Weight (*n* = 89)	Normal Weight (*n* = 432)	*p*	Low-Birth-Weight (*n* = 89)	Normal Weight (*n* = 89)	*p*
Male gender, % (*n*)	55 (49)	61 (56)	0.252	55 (49)	63 (56)	0.287
Age, days	12 (5; 23)	11 (4; 21)	0.800	12 (5; 23)	10 (6; 27)	0.792
Height, cm	46 (43; 48)	52 (51; 53)	<0.001 *	46 (43; 48)	52 (50; 53)	<0.001 *
BSA, m^2^	0.16 (0.15; 0.17)	0.21 (0.20; 0.22)	<0.001 *	0.16 (0.15; 0.17)	0.21 (0.19; 0.22)	<0.001 *
Birth weight, kg	2.05 (1.65; 2.35)	3.29 (2.9; 3.4)	<0.001 *	2.05 (1.65; 2.35)	3.3 (2.86; 3.7)	<0.001 *
Weight at admission, kg	2,2 (1.8; 2.4)	3.35 (2.9; 3.5)	<0.001 *	2.2 (1.8; 2.4)	3.4 (2.9; 3.6)	<0.001 *
Gestational age, weeks	36 [33; 38]	39 [38; 40]	<0.001 *	36 [33; 38]	38.7 [37; 40.5]	<0.001 *
Perinatal encephalopathy, % (*n*)	31 (28)	27 (117)	0.402	31(28)	28 (25)	0.623
Neonatal jaundice, % (*n*)	5 (4)	12 (52)	0.041 *	5 (4)	11 (10)	0.095
MCA, % (*n*)	21 (19)	7 (30)	<0.001 *	21 (19)	26 (23)	0.481
History of infections, % (*n*)	11 (10)	23 (99)	<0.001 *	11 (10)	11 (10)	1.000
PGE1 infusion, % (*n*)	48 (43)	42 (181)	0.321	48 (43)	28 (25)	0.623

BSA—body surface area, MCA—multiple congenital anomalies, PGE1—prostaglandin E1, *—statistically significant differences.

**Table 2 life-13-02282-t002:** Patient parameters according to the initial data.

Parameters	Low-Birth-Weight (*n* = 89)	Normal Weight (*n* = 89)	*p*
Respiratoty Rate, per minute	50 (43; 60)	50 (46; 60)	0.262
Heart Rate, bpm	150 (140; 160)	145 (130; 150)	0.010 *
Creatinine level, mcmol/L	52 (41.5; 68.5)	47.5 (40; 58)	0.113
GFR, ml/min/m^2^	25.1 (18.8; 32.4)	39.5 (31.75; 45.7)	<0.001 *
Lactate, mmol/L	1.8 (1.2; 2.5)	1.8 (1.1; 2.4)	0.701
Z-score of isthmus	−4.3 (−5.65; −3.78)	−5.26 (−6.09; −4.14)	0.012 *
Mean pressure gradient across isthmus, mm Hg	45 (27.7; 55)	48 (30; 62)	0.404
PDA diameter, mm	3 (1.7; 4.7)	3 (1.2; 4.0)	0.385
Z-score of distal arch	−1.1 (−2.1; −0.35)	−0.91 (−2.16; 0.38)	0.047 *
LVEDD Z-score	−0.24 (−1.08; 0,71)	−0.66 (−1.3; 0.47)	0.122
Mitral annulus Z-score	−0.57 (−1.2; 0)	−0.86 [−1.32; −0.14]	0.251
Aortic annulus Z-score	0.48 (−0.24; 1.27)	0.48 (−0.61; 1.19)	0.685
LV ejection fraction, %	63.5 (55; 68)	67 (60; 70)	0.071
Inotropes infusion, % (*n*)	22 (20)	11 (10)	0.045 *
Ross heart failure class:			0.021 *
- I class, % (*n*)	59.5 (53)	75.3 (67)	
- II class, % (*n*)	33.8 (30)	22.5 (20)	
- III class, % (*n*)	6.7 (6)	2.2 (2)	
pRIFLE score:	1 (1; 1)	1 (1; 1)	0.099
- norm, % (*n*)	80 (71)	90 (80)	
- risk, % (*n*)	13.5 (12)	5.6 (5)	
- injury, % (*n*)	5.5 (5)	2.2 (2)	
- failure, % (*n*)	1 (1)	2.2 (2)	
Mechanical ventilation at admission, % (*n*)	21 (19)	5 (4)	<0.001 *
Hepatomegaly, cm	2 (1.5; 2.5)	2 (2; 3)	0.011 *

GFR—glomerular filtration rate, PDA—patent ductus arteriosus, LVEDD—left ventricular end-diastolic dimension, LV—left ventricular, pRIFLE—pediatric RIFLE (acronym indicating risk of renal dysfunction; injury to the kidney; failure of kidney function, loss of kidney function, and end-stage kidney disease), *—statistically significant differences.

**Table 3 life-13-02282-t003:** Intra- and post-operative patients’ data.

Parameters	Low-Birth-Weight (*n* = 89)	Normal Weight (*n* = 89)	*p*
Intra- and postoperative data			
Cross-clamp time, min	19 (15; 24)	20 (15; 25)	0.251
Intraoperative complications, % (*n*)	5.6 (5)	0 (0)	0.012 *
- bleeding, % (*n*)	4.5 (4)	0 (0)	
- thrombosis, % (*n*)	1.1 (1)	0 (0)	
Vasoactive-inotropic score	6.12 (4; 7)	5 (4; 7)	0.942
Post-op arch mean gradient, mm HG	15 (12; 21)	16 (12; 22)	0.414
LV ejection fraction, %	60 (58; 60)	60 (58; 60)	0.134
Lactate 1st day post-op, mmol/L	2.3 (1.7; 4)	2.15 (1.5; 3)	0.064
Lactate 3d day post-op, mmol/L	1.9 (1.4; 2.6)	1.6 (1.2; 2.2)	0.034 *
Creatinine post-op, mcmol/L	62 (47; 90)	55 (49; 65)	0.046 *
GFR post-op, ml/min/m^2^	21.5 (14; 29.2)	34.4 (28.8; 39.4)	<0001 *
pRIFLE, post-op	1 (1; 2)	1 (1; 1)	0.004 *
- norm, % (n)	68.5 (61)	89 (79)	
- risk, % (n)	19 (17)	9 (8)	
- injury, % (n)	4.5 (4)	2 (2)	
- failure, % (n)	8 (7)	0 (0)	
Early postoperative complications			
Infections, % (*n*)	40 (36)	30 (27)	0.108
Pneumonia, % (*n*)	25 (22)	17 (15)	0.144
Necrotising enterocolitis, % (*n*)	5.6 (5)	2.2 (2)	0.009
Acute kidney failure, % (*n*)	5.6 (5)	1.1 (1)	0.054
Systemic inflammatory response syndrome, % (*n*)	8 (7)	9 (8)	0.578
Gastrointestinal bleeding, % (*n*)	2.2 (2)	1.1 (1)	0.313
Atelectasis, % (*n*)	2.2 (2)	4.5 (4)	0.351
Neurological complications, % (*n*)	0 (0)	8 (7)	0.014 *
Outcomes			
In-hospital mortality, % (*n*)	8 (7)	1.1 (1)	0.030 *
Discharged, % (*n*)	25 (22)	82 (73)	<0.001 *
Transferred to other hospitals, % (*n*)	67 (60)	17 (15)	<0.001 *
Gradient before discharge, % (*n*)	15 (10; 20)	15 (10; 20)	0.771

LV—left ventricular, GFR—glomerular filtration rate, pRIFLE—pediatric RIFLE (acronym indicating risk of renal dysfunction; injury to the kidney; failure of kidney function, loss of kidney function and end-stage kidney disease), *—statistically significant differences.

**Table 4 life-13-02282-t004:** Uni- and multivariate logistic regression analysis in patients for the outcome of coarctation.

Parameters	Univariate Regression Analysis OR (95% CI)	*p*	Multivariate Regression Analysis OR (95% CI)	*p*
Respiratory rate, per minute	1.061 (1.013–1.111)	0.011 *	-	-
Inotropes infusion	4.285 (1.438–12.77)	0.012 *	4.369 (1.316–14.51)	0.016 *
Ross heart failure class	2.539 (1.231–5.235)	0.014 *	-	-
Cross-clamp time, min	0.897 (0.825–0.974)	0.004 *	0.922 (0.845–1.001)	0.054
Gradient before discharge, mmHG	1.084 (1.022–1.196)	0.006 *	1.081 (1.014–1.153)	0.016 *

OR—odds ratio, CI—confidence interval, *—statistically significant differences.

## Data Availability

The primary data analysed in this study are not publicly available due to the policy of access to clinical data of the Bakulev Center for Cardiovascular Surgery of the Ministry of Health of the Russian Federation. However, some parameters that do not contain personal information can be provided by the corresponding author upon reasonable request.

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
