# Peer review of "Results of Aortic Coarctation Repair in Low- and Normal Birth-Weight Neonates: A Propensity Score-Matched Analysis"

_life, 2023, doi:10.3390/life13122282_

Round 1

Reviewer 1 Report

Comments and Suggestions for Authors

Dear authors,

Coarctation of the aorta is a serious cardiac malformation and its treatment in a premature newborn with low birth weight is a challenge for the cardiovascular surgeon. The chosen subject is an important one and deserves to be studied, and the results of your study deserve to be published after reviewing certain aspects.

Abstract - it has to be reformulated. Too many repeating of Low-birth weight word. Once you have established the abbreviation LBW for low-birth weight, it must be used in the following sentences.

Introduction : -Same remark as above. 

Methods and Results: - Well argued;

- * — statistically significant differences: this is redundant. p<0.05 is statistically significant, you do not need to mention it every time.

Discussion: Please add a Limitation of the study chapter.

Comments on the Quality of English Language

The manuscript must be reviewed by a native English speaker.

Author Response

We are grateful to our reviewer for his or her very important comments to our work. We tried to correct all shortcomings according to these comments. All changes in the text are highlighted in green.

Coarctation of the aorta is a serious cardiac malformation and its treatment in a premature newborn with low birth weight is a challenge for the cardiovascular surgeon. The chosen subject is an important one and deserves to be studied, and the results of your study deserve to be published after reviewing certain aspects.

Abstract - it has to be reformulated. Too many repeating of Low-birth weight word. Once you have established the abbreviation LBW for low-birth weight, it must be used in the following sentences.

We fixed it.

Introduction : -Same remark as above. 

We fixed it.

Methods and Results: - Well argued;

- * — statistically significant differences: this is redundant. p<0.05 is statistically significant, you do not need to mention it every time.

We fixed it.

Discussion: Please add a Limitation of the study chapter.

We have added Limitations.

Limitations

This analysis had both advantages and limitations, primarily related to its retrospective approach. Firstly, despite the fact that all data were collected from the electronic database of our clinic “Medwork” with standard mandatory data entry, retrospective analysis does not exclude partial loss of information. Secondly, the use of propensity score matching does not completely exclude hidden systematic error, which could potentially affect the results. Thirdly, long-term data were not collected from all patients: 63 children in the study group and 76 patients in the control group.

We assume that the advantages of this study include a relatively large sample size including 89 LBW neonates with aortic coarctation, as well as a long follow-up period: the average duration was 15 years. For the first time in our country based on a large clinical material a comparative assessment of LBW neonates with aortic coarctation during all stages of treatment was carried out, risk factors for recoarctation were identified.

Reviewer 2 Report

Comments and Suggestions for Authors

This paper compares the prognosis of surgery for coarctation of the aorta between low-birth-weight and normal-birth-weight infants, using the Propensity score to match factors other than weight and week of gestation, and is considered to be a valuable report.

However, the following points may need to be addressed.

Major

1. What are the criteria for indication for surgery? If the indication differs between the two groups, the prognosis may also differ, so even if there is no difference, it may be necessary to describe it.

2. The LBW group seems to have a significantly earlier weeks of gestation.Is it possible that this is the reason for the difference in heart rate and renal function compared to normal-weight infants? It would be better to mention in the discussion whether this is due to aortic stenosis or preterm delivery.

Minor

The table on page 7 is not titled.

Author Response

We are grateful to our reviewer for his or her very important comments to our work. We tried to correct all shortcomings according to these comments. All changes in the text are highlighted in green.

Major

1. What are the criteria for indication for surgery? If the indication differs between the two groups, the prognosis may also differ, so even if there is no difference, it may be necessary to describe it.

Indications for surgery did not differ between studied groups and were following: isthmus Z-score less than -3, pressure gradient more than 20 mmHg and symptoms of heart failure based on Ross Classification.

We have added this information to the manuscript.

2. The LBW group seems to have a significantly earlier weeks of gestation.Is it possible that this is the reason for the difference in heart rate and renal function compared to normal-weight infants? It would be better to mention in the discussion whether this is due to aortic stenosis or preterm delivery.

Yes, there is a possibility that heart rate and impaired renal function might be due to early gestational age in patients with mild symptoms of heart failure (Ross class I), but among infants with initially severe clinical condition we assume that these parameters are mostly the manifestation of heart failure.

Minor

The table on page 7 is not titled.

We fixed it.

Reviewer 3 Report

Comments and Suggestions for Authors

This article by Krylova et al analyzes in hospital postoperative mortality and long term survival (both disease free and with recoarctation) in a group of low birth weight infants compared to normal weight infants.

How was the diagnosis of coarctation established? By echocardiography or angio CT? What were the severity classes of the coarctation in terms of gradients?

How many newborns had their PDA open at the time of the intervention? Can you elaborate on how a PDA may impair coarctation severity estimation by echocardiography?

Were there any cases of interrupted aortic arch?

How soon after surgery were the patients discharged home? The question also refers to the duration of hospitalization in the centers where the LBW infants were transferred. Are there any significant differences between the 2 groups?

The paper only has 27 references.

Author Response

We are grateful to our reviewer for his or her very important comments to our work. We tried to correct all shortcomings according to these comments. All changes in the text are highlighted in green.

This article by Krylova et al analyzes in hospital postoperative mortality and long term survival (both disease free and with recoarctation) in a group of low birth weight infants compared to normal weight infants.

How was the diagnosis of coarctation established? By echocardiography or angio CT? What were the severity classes of the coarctation in terms of gradients?

Coarctation was diagnosed by echocardiography, patients with hypoplastic aortic arch or unusual anatomy underwent CT scan before surgery. We did not access severity of coarctation based only on pressure gradient, we always assessed comprehensively echocardiographic data, clinical condition and severity of heart failure.

We have added this information to the manuscript.

How many newborns had their PDA open at the time of the intervention? Can you elaborate on how a PDA may impair coarctation severity estimation by echocardiography?

In normal-weight group 89 patients had functioning PDA, in LBW – 87 patients. In our study PDA diameter did not influence the mortality rates, only 2 patients with closed PDA in LBW group who were clinically stable upon admission. PDA diameter does not always show the severity of coarctation.

Were there any cases of interrupted aortic arch?

No, we selected patients only with coarctation of the aorta.

How soon after surgery were the patients discharged home? The question also refers to the duration of hospitalization in the centers where the LBW infants were transferred. Are there any significant differences between the 2 groups?

Patients in both groups who were discharged home had approximately the same duration of hospitalization - about 8-10 days. Many patients were transferred to our center only for surgical treatment for 2-3 days due to legal reasons and then transferred back to continue treatment in specialized perinatal centers, so we did not analyze the length of hospital stay. We were unable to collect data on patients who transferred to other hospitals because of technical reasons, so we described the outcomes of hospitalizations: number of patients who were discharged home, transferred patients, and deaths.

We have added this information to the manuscript.

The paper only has 27 references.

We have added a few more references.

Round 2

Reviewer 1 Report

Comments and Suggestions for Authors

Dear authors,

Your manuscript is improved and may be published.

Reviewer 2 Report

Comments and Suggestions for Authors

It is now easier to understand.

It is acceptable for publication.

Reviewer 3 Report

Comments and Suggestions for Authors

The authors have adequately addressed my concerns. Thank you!